## Research Article

adolescents and young adults; alcohol and substance use; depression and anxiety; informal settlements; mental health

**Corresponding author:**
Catherine Mawia Musyoka;
Email: camulundu2011@gmail.com

# A pilot study of alcohol and substance use, mental health symptoms and social vulnerabilities among youth in Nairobi's informal settlements

Catherine Mawia Musyoka[1] , William Byansi[2], Teresia Mutavi[1], Anne Mbwayo[1] ,
Dorothy Ndunge Kyalo[3], Angeline Mulwa[3], Sungseek Moon[4] and Muthoni Mathai[1]

[1]Department of Psychiatry, Faculty of Health Sciences, University of Nairobi, Kenya; [2]Social Work, Boston College, USA; [3]Education, University of Nairobi College of Education and External Studies, Kenya and [4]Diana R. Garland School of Social Work, Baylor University, USA

## Abstract

Alcohol and Substance Use (ASU) and mental ill-health among youths is today a global public health concern especially among the urban poor. This pilot study examined the prevalence, patterns and mental health associations of ASU among youths in urban slums. Baseline cross-section data were collected from 94 participants aged 15–24 in two informal settlements in Nairobi. Descriptive statistics analyzed demographic, substance use and mental health variables. Bivariate analyses of associations between ASU scores, sociodemographic factors and mental health symptoms were done. Seventy-eight per cent of participants reported having used alcohol in the preceding 3 months, while 68% and 35% respectively reported cannabis and tobacco use. Concerning frequency of use, 43% used alcohol while 47% used cannabis frequently. Alcohol use was associated with age, depressive symptoms and socio-economic independence. Tobacco use was more common among participants with depression, anxiety and low education levels. Cannabis use was higher in participants living independently, with depression, anxiety and stress and in men. In conclusion the study found prevalent ASU associated with multiple sociodemographic and psychological vulnerabilities. These findings may reflect sample characteristics not generalizable to the population, but they provide preliminary evidence for the need of future studies of integrated preventive interventions.

## Impact statement

This pilot study provides early evidence about alcohol, tobacco and cannabis use and mental health challenges among adolescents and youth living in informal settlements in Nairobi, Kenya. It explored how frequently youth reported using substances like alcohol, tobacco and cannabis and how these behaviors are associated with symptoms of depression, anxiety and stress. Because of the small, nonrandom sample, the findings are not meant to represent all young people in Nairobi's informal settlements. Instead, they offer important initial insights that can help shape future research and guide early prevention efforts. We found that alcohol, tobacco and cannabis use were common among this group, and many young people reported using these substances weekly or daily. Substance use was associated with several factors, including age, gender, living arrangement, education and employment status. Young people with elevated symptoms of depression, anxiety and stress were more likely to use substances. Those living independently or working were more likely to use alcohol and cannabis, while tobacco use was more common among those with lower levels of education. These exploratory findings highlight how substance use and mental health are interconnected and shaped by social and economic vulnerabilities. Pilot results suggest that prevention and treatment programs need to address both substance use and mental health simultaneously. Because this was an exploratory pilot study, more research with larger and more representative samples is needed to understand these issues better. The results can help researchers, policymakers and community organizations develop interventions that are relevant, practical and responsive to the needs of young people living in urban informal settlements.

## Introduction

Alcohol and substance use (ASU) among adolescents and young adults is a growing public health concern globally, with a particularly significant impact in sub-Saharan Africa. The developmental period between the ages of 15 and 24 is marked by rapid biological, psychological and social

transitions that increase vulnerability to risk behaviors, including substance use (Arnett, 2000; Kessler et al., 2007). This period often coincides with heightened experimentation, peer influence and exposure to stressors such as unemployment or limited educational opportunities. Importantly, this is also the life stage during which many mental health disorders first emerge, creating a critical intersection between substance use and mental health (Arnett, 2000; Kessler et al., 2007). When substance use occurs alongside these vulnerabilities, it can exacerbate the risks of long-term adverse outcomes, including dependence, poor educational attainment and reduced economic productivity (Mangeni and Mbuthia, 2018; Kumar et al., 2024). It is essential to recognize and address ASU during this critical developmental stage to protect the immediate well-being and prospects of adolescents and young adults. This pilot study is exploratory in nature and aims to generate preliminary insights to inform future, larger studies using more representative designs.

Globally, mental health conditions are among the leading causes of disease burden, with approximately 75% of all mental health disorders beginning during adolescence and early adulthood (Kessler et al., 2007; Mokdad et al., 2016). A recent Lancet study reported that the average age of onset for mental health disorders is 15 years (McGrath et al., 2023), a time when many young people also begin experimenting with alcohol and other substances.

ASU are closely intertwined with the onset of common mental health symptoms such as depression, anxiety and stress. Among youth aged 15–19, depression and anxiety-related conditions rank among the top five contributors to years lived with disability (Mokdad et al., 2016), and nearly one-quarter of disability-adjusted life years in this age group are attributed to mental and substance use disorders (Whiteford et al., 2013). These statistics underscore the need to understand and address the co-occurrence of substance use and mental health challenges among adolescents and young adults, particularly in resource-limited contexts where prevention and treatment services are scarce. Failure to intervene during adolescence not only increases the risk of acute harms, such as self-harm and premature mortality, but also sets the stage for chronic health, social and economic disadvantages across the life course.

In Kenya, the burden of substance use and mental health issues is worsened by rapid urbanization and ongoing social inequalities. Nairobi, the capital city, is home to more than half of the population, who live in informal, densely packed settlements characterized by extreme poverty, high unemployment, limited access to clean water and sanitation and insufficient health and education services (Odongo and Donghui, 2021; Kibichii and Mwaeke, 2024). These environments contribute to chronic stress, insecurity and lack of opportunity, which, in turn, increase the risk of substance use and mental health problems among young residents (Ndugwa et al., 2011; Kibichii and Mwaeke, 2024). Despite the magnitude of the issue, research focused on adolescents and young adults in Nairobi's informal settlements remains limited.

This pilot study seeks to fill key gaps in the literature by examining the prevalence of ASU and associated mental health symptoms among youth aged 15–24 years in Nairobi's informal settlements. First, it contributes to our understanding of the prevalence and co-occurrence of mental health symptoms and substance use among a sample of vulnerable youth in urban poor contexts. Second, it explores demographic and contextual risk and protective factors that may inform tailored prevention strategies. These factors include gender, education and housing conditions associated with substance use and mental health symptoms. Third, it provides

empirical evidence to support context-specific interventions and policy efforts aimed at reducing youth substance use and improving mental health outcomes in informal settlements. Therefore, this study aims to assess the prevalence and patterns of ASU and their associated mental health symptoms among adolescents and young adults aged 15–24 years living in Nairobi's informal settlements.

## Materials and methods

### Study design and setting

We used baseline data from a cluster-randomized mixed-methods sequential study conducted in two informal settlements in Nairobi, Kenya. The study involved a community-based sample of youth from two informal settings in the Nairobi Metropolitan area, rather than a clinical or patient population. The present manuscript reports only the baseline data collected between September and December 2024 prior to randomization. Qualitative interviews were conducted later within the intervention community to document user experience, and those findings are outside the scope of this paper.

### Study participants

This pilot study used a snowball sampling method to recruit 94 youth aged 15–24 years. Recruitment was supported by community health promoters (CHPs), who are nonprofessional youth leaders living within the community. Because CHPs work and interact daily with residents, they are familiar with adolescents' and youths' behaviors, including ASU. Using this knowledge, CHPs identified three initial participants (two males and one female), who then referred peers with similar behaviors for recruitment. While this approach facilitated access to adolescents and youths using alcohol, it may also have introduced bias, as CHPs' perceptions and social networks could influence which youths were included.

Eligibility criteria included being between 15 and 24 years of age, having resided in the study location for at least 3 months prior to study commencement and providing informed consent to participate in the behavioral health screening survey. For participants aged 15–17 years, parental or guardian consent and participant assent were required. Eligible participants also needed to have a score below 15 on the alcohol, smoking, and substance involvement screening test (ASSIST), indicating mild to moderate substance use and access to a mobile phone for study follow-up. Both in-school and out-of-school youth were included, as many in this age group were not attending school due to economic hardship or delinquent behaviors common in the setting.

Exclusion criteria included youth outside the specified age range, those who had lived in the community for <3 months, and individuals with severe substance use disorder (ASSIST score ≥ 15) or mental health symptoms requiring inpatient management. In addition, youth who were only visiting the study areas and were unlikely to remain for the full study period were excluded. Those identified with severe substance use problems were referred to government hospitals for appropriate care.

To uphold confidentiality, all study personnel completed refresher training in human subject's protection and Good Clinical Practice through the NIDA Clinical Trials Network. Study staff also received in-person training on maintaining confidentiality, and all documentation was handled with the strictest care. Parents and guardians were not permitted to sit in on their adolescents' study proceedings to protect privacy. For participants under 18 years of

age, adequate provisions were made to obtain both parental/guardian permission and youth assent. Participants themselves provided assent and were explicitly told they could withdraw at any point without penalty. They were also informed that parents or guardians would not be notified of their responses unless there was a life-threatening situation or reported child abuse. Together, these safeguards ensured that the study was conducted following ethical standards and that the rights and confidentiality of participants were fully protected.

## Procedures

### Research assistant section and training

Five research assistants (RAs) were recruited through a competitive hiring process following a public call for applications. All five RAs held at least a bachelor's degree in public health, psychology, or a related field. Candidates were evaluated based on their prior research experience, familiarity with working in community settings and interpersonal skills during structured interviews. The selected RAs completed an online certification in Good Clinical Practice conducted by the NIDA Clinical Trials Network, which emphasized the principles of ethical research conduct. In addition, RAs received in-person training from the PI and Co-I tailored to the study context, which included good data collection techniques, ensuring confidentiality, mandatory reporting procedures and strategies for managing sensitive topics such as substance use and mental health. This combination of formal certification and study-specific training ensured that the research team was well prepared to collect high-quality data while upholding ethical standards.

### Participant recruitment

Researchers collaborated with two youth-serving organizations, Mtaa Safi in Mukuru Kwa Reuben and Kibra Youth from Kibra informal settlement, to support the recruitment of participants. RAs and CHPs distributed flyers with a short study description and information on the time and location of data collection. Once the initial seed of participants was identified, each youth who enrolled was encouraged to invite up to four from their networks who also fit the eligibility criteria. This peer referral approach was designed to expand the sample and reach hidden participants, particularly youth who might not be directly connected to formal services but were known through social networks.

### Data collection

Written informed consent (and assent for minors) was obtained from all participants, as well as from parents and caregivers of those under 18, with separate consent sought to avoid coercion. Eligibility screening was conducted in person, and ineligible individuals were excluded. Trained RAs fluent in English and Swahili conducted the interviews in the participant's preferred language. Surveys took about 30 minutes and were administered in private to ensure confidentiality.

### Ethics

The study received ethical approval from the Kenyatta National Hospital and University of Nairobi (KNH-UoN ERC) No. P423/05/2024 and the National Commission for Science, Technology & Innovation (NACOSTI) No. NACOSTI/P/24/39331. Informed written consent/assent was obtained from participants before data collection commenced. For participants younger than 18, consent to participate was obtained from their parents or legal guardians. Participants identified with severe substance use problems were referred to government hospitals for appropriate care. Participants screening positive for probable depression (Patient Health Questionnaire-9 [PHQ-9] ≥ 10) or probable anxiety (generalized anxiety disorder-7 [GAD-7] ≥ 8) received a brief resource sheet listing local primary care and mental health services. Any endorsement of suicidality on the PHQ-9 triggered an immediate risk assessment by a trained RA under the PI's (psychologist) oversight, safety planning, notification of the legally authorized representative when appropriate and facilitated referral to government health facilities in Nairobi per our IRB-approved crisis protocol. Participants received 500 Kenya shillings (approximately 5 USD) as compensation for transport costs. Their participation was therefore totally voluntary.

## Measures

The study utilized standardized, validated instruments to assess substance use and mental health outcomes, along with key demographic and contextual variables. Specifically, the Patient Health Questionnaire-9 (PHQ-9) and GAD-7 have both been adapted and validated for use in Kenya in English and Swahili (Osborn et al., 2021; Odero et al., 2023; Tele et al., 2023). The PHQ-9 has demonstrated reliability and validity among adolescents and adults (Osborn et al., 2021).

### Tobacco, alcohol and cannabis use (dependent outcomes)

We assessed substance use with the ASSIST, an eight-item instrument developed by the World Health Organization (WHO-ASSIST-Working-Group, 2002) to assess lifetime and recent use of alcohol and other substances. Because the original ASSIST was designed for adults, a modified version (ASSIST-Y) has been validated for use with adolescents aged 15–17. ASSIST-Y omits item 7, which asks about failed attempts to reduce substance use. To standardize the tool, we administered ASSIST-Y to all participants, regardless of age, and calculated both frequency of use and cumulative scores per substance within the last 3 months, consistent with ASSIST-Y methodology. We focus on tobacco, alcohol and cannabis, which were the most widely used substances in the prior 3 months; fewer than 10% of participants reported use of any other individual substance during that period. Possible tobacco scores ranged from 0 to 25, and alcohol and cannabis scores ranged from 0 to 33. The scale has been validated in several low-resource settings, including India, Brazil (Humeniuk et al., 2012) and Kenya (Jaguga et al., 2023). We did not classify ASSIST scores into risk categories (low/moderate/high), given the sample size. Scores are analyzed continuously and as frequencies.

### Depression

The Patient Health Questionnaire-9 (PHQ-9) (Kroenke et al., 2001) is a self-report tool designed to assess depressive symptoms, including feelings of hopelessness, sleep disturbances and negative thoughts. It has been widely used in adolescent populations (Richardson et al., 2010) and is effective for tracking changes in depressive symptoms over time (Löwe et al., 2004). The PHQ-9 consists of nine items rated on a 4-point scale, ranging from 0 (not at all) to 3 (nearly every day). The cumulative scores can range from 0 to 27, indicating varying levels of depression severity. Interpretation of scores is as follows: minimal depression (0–4),

mild depression (5–9), moderate depression (10–14), moderately severe (15–19), or severe (20–27). This tool has good psychometric properties and has been validated and translated into the Swahili language in Kenya (Osborn et al., 2021; Odero et al., 2023; Tele et al., 2023) and used extensively in many culturally diverse countries (Kochhar et al., 2007). It has good internal consistency in the current study (Cronbach's α = 0.80), and we used a score of 10 or above as an acceptable measure of probable depression.

### Anxiety

The Generalized Anxiety Disorder-7 (GAD-7) (Spitzer et al., 2006) is a 7-item self-report scale measuring anxiety symptoms over the past 2 weeks. Items are rated from 0 (not at all) to 3 (nearly every day), yielding a total score between 0 and 21. Severity levels are: minimal (0–4), mild (5–9), moderate (10–14) and severe (15–21). The GAD-7 has been widely validated across settings (Spitzer et al., 2006; Jordan et al., 2017), including in Kenya (Osborn et al., 2021; Odero et al., 2023). A score of 8 and above was considered positive for probable anxiety. The GAD-7 demonstrated acceptable internal consistency in the current study (Cronbach's α = 0.81).

### Stress

The perceived stress scale (PSS) (Cohen et al., 1983) is a widely used self-report tool that assesses how individuals perceive stress in their daily lives, focusing on feelings of unpredictability, lack of control and overload. Items are rated on a 5-point Likert scale (0 = never to 4 = very often). Total scores range from 0 to 40, with higher scores indicating greater perceived stress. The PSS has been validated across diverse populations and is commonly used to examine the impact of stress on mental and physical health outcomes (Manzar et al., 2019). A score of 27 and above was considered probable severe stress. In this study, the scale demonstrated acceptable internal consistency (Cronbach's α = 0.79). In Kenya, the scale has demonstrated good reliability and validity (Afulani et al., 2021).

### Sociodemographic variables

Included age, gender (male vs. female), education level (less than secondary vs. higher than secondary), occupation (working vs. not working) and current residence (parental or guardian care vs. independent living).

### Analytical approach

Variables were summarized using descriptive statistics appropriate to their level of measurement and distribution. We report the frequency of tobacco, alcohol and cannabis, along with measures of central tendency and variability overall and stratified by sociodemographic and mental health variables. We conducted bivariate analyses between each outcome and sociodemographic and mental health factors. For alcohol, which was normally distributed, we used Pearson correlation for the continuous age variable, independent-samples *t*-tests for the binary variables and analysis of variance (ANOVA) for the categorical age variable. For tobacco and cannabis, which were not normally distributed, we conducted Spearman correlations, Wilcoxon rank-sum tests and Kruskal–Wallis tests.

Analyses were limited to descriptive and bivariate statistics to examine associations between sociodemographic characteristics; alcohol, tobacco and cannabis use; and mental health symptoms (depressive, anxiety and stress symptoms). Given the modest sample size (*N* = 94), conducting multivariate analyses would risk unstable estimates and overfitting, thereby reducing the reliability of results. Bivariate analyses were therefore deemed most appropriate for this pilot investigation, which was designed to provide preliminary insights rather than definitive causal inferences.

## Results

The median age of participants was 21 years (interquartile range [IQR] = 4), with a little over half of the sample being male (54%; *n* = 51) (Table 1). Most participants (81%, *n* = 76) had less than a secondary education, while 19% (*n* = 18) had higher education, including college and university. Additionally, 70% (*n* = 66) lived under parental or guardian care, and 64% (*n* = 60) were unemployed. Participants reported an average stress symptom score of 19.08 (standard deviation [SD] = 7.51). The median depressive symptoms score was 5 (IQR = 8), and the median anxiety symptoms score was 7 (IQR = 7). For probable clinical conditions, 25% (*n* = 24) of participants met criteria for probable depression, 34% (*n* = 32) met criteria for probable anxiety and 14% (*n* = 13) met criteria for probable stress. Within the prior 3 months, most participants reported alcohol use (78%; *n* = 73) or cannabis use (68%; *n* = 63), while 35% (*n* = 33) reported tobacco use (Table 2). Nearly half of participants reported frequent use, with 43% (*n* = 37) using alcohol and 47% (*n* = 44) using cannabis weekly, daily, or almost daily.

Bivariate analyses revealed that higher depression and anxiety scores, whether measured continuously or categorically, were consistently associated with greater tobacco, alcohol and cannabis scores, while stress was only associated with cannabis use (Table 1). Participants with less than secondary education had significantly higher tobacco use scores than those with higher education (*z* = 2.48, *p* = 0.005), despite the same median, reflecting greater variability and higher values in the lower education group. Alcohol scores increased with age (ρ = 0.40, *p* < .001) and were higher among those who were working versus not working (*t*(91) = −2.25, *p* = 0.027) and those living independently versus with parents/guardians (*t*(91) = −2.88, *p* = 0.005). Cannabis scores were higher for males (*z* = −2.52, *p* = 0.011) and those living independently (*z* = −2.51, *p* = 0.012) compared with their counterparts.

## Discussions

This pilot study provides insights into the patterns and correlates of ASU as well as associated mental health symptoms among adolescents and young people living in informal settlements in Nairobi, Kenya. Within this sample, a high proportion reported alcohol, tobacco and cannabis use, alongside notable levels of probable depression, anxiety and stress symptoms. Substance use was significantly associated with sociodemographic and mental health factors such as age, education level, employment status and living arrangements. Tobacco use was elevated among those with higher depression and anxiety symptoms and lower educational attainment. Alcohol use was linked with older age, higher depressive symptoms and independent living or employment. Cannabis use was higher among males; those living independently; and those with elevated depression, anxiety and stress symptoms. These exploratory findings underscore the complex interplay between sociodemographic vulnerabilities, mental health and substance use within this vulnerable population. Although the measures of mental health (Afulani et al., 2021; Osborn et al., 2021; Tele et al., 2023) and substance use (Jaguga et al., 2023) have been validated for use among Kenyan adolescents, continuous attention to their

**Table 1.** Descriptive statistics and bivariate tests of association

| | N (%), median (IQR), or mean (SD) | Tobacco n = 91 | | | Alcohol n = 93 | | | Cannabis n = 93 | | |
|---|---|---|---|---|---|---|---|---|---|---|
| | | Median (IQR) | Test statistic | p | Mean (SD) | Test statistic | p | Median (IQR) | Test statistic | p |
| Total sample | 94 | 0.0 (3.0) | | | 13.6 (9.9) | | | 13.0 (21.0) | | |
| Age | 21 (4) | – | ρ = −0.07 | 0.504 | – | ρ = 0.40 | 0.000 | – | ρ = 0.11 | 0.292 |
| Age | | | $\chi^2$ = 3.20 | 0.362 | | F = 5.86 | 0.001 | | $\chi^2$ = 1.39 | 0.709 |
| 15–17 | 16 (17.0) | 0.0 (10.0) | | | 8.5 (8.4) | | | 9.0 (20.0) | | |
| 18–19 | 24 (25.5) | 0.0 (1.0) | | | 9.6 (9.5) | | | 10.0 (18.5) | | |
| 20–22 | 31 (33.0) | 0.0 (0.0) | | | 15.2 (8.9) | | | 14.0 (21.0) | | |
| 23–25 | 23 (24.5) | 0.0 (5.0) | | | 18.8 (9.6) | | | 17.0 (25.0) | | |
| Sex | | | z = −0.34 | 0.746 | | t = 0.00 | 0.999 | | z = −2.52 | 0.011 |
| Female | 43 (45.7) | 0.0 (2.0) | | | 13.6 (10.3) | | | 2.0 (20.0) | | |
| Male | 51 (54.3) | 0.0 (5.0) | | | 13.6 (9.6) | | | 15.5 (14.0) | | |
| Education | | | z = 2.48 | 0.005 | | t = −1.06 | 0.291 | | z = 1.46 | 0.144 |
| Secondary or less | 76 (80.9) | 0.0 (7.0) | | | 13.0 (9.8) | | | 13.0 (22.0) | | |
| Higher than secondary | 18 (19.2) | 0.0 (0.0) | | | 15.8 (10.3) | | | 0.0 (21.0) | | |
| Employment | | | z = −0.61 | 0.542 | | t = −2.25 | 0.027 | | z = −1.65 | 0.100 |
| Not working | 60 (63.8) | 0.0 (2.0) | | | 11.8 (9.5) | | | 9.0 (20.0) | | |
| Working | 34 (36.2) | 0.0 (7.0) | | | 16.5 (9.9) | | | 15.0 (20.0) | | |
| Living situation | | | z = −0.30 | 0.755 | | t = −2.88 | 0.005 | | z = −2.51 | 0.012 |
| With parent/guardian | 66 (70.2) | 0.0 (2.0) | | | 11.7 (9.3) | | | 7.0 (20.0) | | |
| Alone or with spouse | 28 (29.8) | 0.0 (5.0) | | | 17.9 (10.0) | | | 18.0 (13.5) | | |
| Probable depression | | | z = −2.60 | 0.012 | | t = −2.14 | 0.035 | | z = −2.34 | 0.019 |
| No | 70 (74.5) | 0.0 (0.0) | | | 12.0 (16.0) | | | 11.0 (18.0) | | |
| Yes | 24 (25.5) | 1.0 (9.5) | | | 20.0 (17.0) | | | 20.0 (23.0) | | |
| Probable anxiety | | | z = −3.45 | 0.000 | | t = −2.07 | 0.041 | | z = −3.00 | 0.003 |
| No | 62 (66.0) | 0.0 (0.0) | | | 12.0 (15.0) | | | 8.0 (18.0) | | |
| Yes | 32 (34.0) | 2.0 (13.0) | | | 18.0 (17.0) | | | 20.0 (20.0) | | |
| Probable stress | | | z = −0.23 | 0.797 | | t = −0.99 | 0.325 | | z = −2.09 | 0.037 |
| No | 81 (86.2) | 0.0 (3.0) | | | 12.0 (18.0) | | | 11.5 (20.0) | | |
| Yes | 13 (13.8) | 0.0 (3.0) | | | 15.0 (22.0) | | | 20.0 (17.0) | | |
| Depression score | 5 (8) | – | ρ = 0.30 | 0.004 | – | ρ = 0.25 | 0.015 | – | ρ = 0.40 | 0.000 |
| Anxiety score | 7 (7) | – | ρ = 0.33 | 0.002 | – | ρ = 0.19 | 0.069 | – | ρ = 0.39 | 0.000 |
| Stress score | 19.1 (7.5) | – | ρ = −0.10 | 0.364 | – | r = 0.13 | 0.201 | – | ρ = 0.21 | 0.046 |

IQR, interquartile range; SD, standard deviation.
Pearson correlation was performed for alcohol with stress; Spearman correlations for all other continuous variable combinations. Independent-samples t-tests were used for alcohol by binary variables, Wilcoxon rank-sum tests for tobacco and cannabis by binary variables, Kruskal–Wallis tests for tobacco and cannabis by the categorical age variable and ANOVA for alcohol by the categorical age variable.
Tobacco was measured on a 0–25 scale and alcohol and cannabis on a 0–33 scale. Depression, anxiety and stress were measured on 0–27, 0–21 and 0–40 scales, respectively. Higher scores indicate greater risk or symptom severity.

cultural relevance remains crucial, especially in informal settlement contexts where distress may be expressed through somatic or culture-specific idioms. Future research should examine the sensitivity of these tools to locally relevant symptoms and consider combining standardized tools with qualitative methods to better understand the lived experiences of mental health among youth in urban informal settlements. It is important to note that ASSIST-Y captures use in the past 3 months; our prevalence estimates might overstate ongoing or dependent use. Findings should be viewed as indicators of recent involvement rather than clinical diagnoses or polysubstance dependence.

These exploratory findings must be interpreted within the context of Nairobi's informal settlements, which are shaped by poverty, unemployment, overcrowding and fragile health and educational

**Table 2.** Reported frequency of use in the past 3 months for tobacco, alcohol and cannabis

|  | Tobacco | Alcohol | Cannabis |
|---|---|---|---|
|  | *n* (%) | *n* (%) | *n* (%) |
| Never | 70 (75.3) | 20 (21.5) | 30 (32.3) |
| Once or twice | 5 (5.4) | 16 (17.2) | 10 (10.8) |
| Monthly | 6 (6.5) | 20 (21.5) | 9 (9.7) |
| Weekly | 9 (9.7) | 32 (34.4) | 25 (26.9) |
| Daily or almost daily | 3 (3.2) | 5 (5.4) | 19 (20.4) |
| *n* | 93 | 93 | 93 |

infrastructures (Stenton, 2015; Odongo and Donghui, 2021; Mukolwe et al., 2024). These structural factors both increase exposure to psychosocial stressors and constrain access to protective resources, thereby shaping patterns of substance use and mental health outcomes. For instance, unemployment was associated with higher alcohol use, likely reflecting heightened stress and limited opportunities. At the same time, employment and independent living were associated with increased alcohol and cannabis use, perhaps due to greater income and autonomy facilitating access. Similarly, lower educational attainment was associated with higher tobacco use, consistent with evidence that schooling provides not only academic instruction but also protective social and health-related knowledge (Lee et al., 2015; Mutai et al., 2020). At the same time, recent evidence of high substance use among university students (23.4%) indicates that education alone is not sufficient protection, emphasizing the need for prevention strategies across all educational levels (Musyoka et al., 2020; Kamenderi et al., 2025).

We also found that youth living independently reported higher alcohol and cannabis use than those living with parents or guardians. While parental co-residence is often assumed to be protective, independent living in this context may provide greater freedom, income control and social exposure that facilitate substance use. Conversely, the protective potential of co-residence may be undermined by factors such as overcrowding, family conflict and parental substance use, which are common in informal settlements (Lee et al., 2015; Mutai et al., 2020; Makokha et al., 2021). These findings point to the dual role of family and household settings as both protective and risk-laden environments, highlighting the need for nuanced family and household-level interventions.

The association between substance use and mental health symptoms, including depression, anxiety and stress, is consistent with existing literature on the bidirectional relationship between these factors (Moustafa et al., 2022; Lazzaro, 2023). For example, higher depression and anxiety were associated with greater tobacco and alcohol use, while higher stress, depression and anxiety were all associated with cannabis use. Given the cross-sectional nature of this study, causal direction cannot be established. Nonetheless, the co-occurrence of substance use and mental health challenges underscores the need for integrated, contextually appropriate interventions that address both substance use and mental health, simultaneously, rather than as separate issues.

Taken together, these exploratory findings highlight the need to understand substance use within the intersecting structural and individual realities of Nairobi's informal settlements. Prevention and intervention efforts should not only focus on individual behaviors but also address systemic factors such as unemployment, poor housing conditions and limited educational opportunities.

Strategies may include school-based prevention programs, youth employment and skills-building initiatives, integration of substance use and mental health services into primary care, and community outreach that incorporates harm reduction. Barriers such as stigma, limited resources and weak referral systems must be anticipated and solutions codesigned with youth and local stakeholders to ensure relevance and sustainability.

### *Limitations and recommendations*

The study has several limitations. First, its cross-sectional design prevents any conclusions about causality or the sequence of events between substance use and mental health symptoms. Therefore, we describe associations rather than causal relationships. Second, the snowball sampling introduced the possibility of selection bias and limited the representativeness of the sample. Recruitment relied on peer networks, and as such, the most marginalized or socially isolated youth may have been underrepresented. Furthermore, the small sample size (*n* = 94) drawn from only two informal settlements further constrains the generalizability of findings. The use of snowball sampling likely introduced selection bias. Because recruitment began through CHPs and relied on peer referral, the sample may overrepresent youth who are more socially connected or embedded within networks where ASU are common. Consequently, the reported prevalence rates may overestimate substance use compared to less connected or more isolated youth in these settings. Additionally, the reliance on self-report may have introduced recall or social desirability biases. The sample included both in-school and out-of-school youth, providing some diversity in experiences. Therefore, the results should be interpreted with caution and viewed as exploratory.

Finally, a key limitation is that analyses were restricted to bivariate associations. Although this approach was appropriate given the small sample size and exploratory nature of the study, it limited our ability to adjust for potential confounding variables. As such, the associations reported may not fully account for these influences. Future studies should use longitudinal designs with probability sampling approaches and recruit across multiple informal settlements within Nairobi and other Kenyan cities to improve external validity and capture neighborhood-level heterogeneity. Additionally, future research should explore the roles of family dynamics, parental substance use, community violence and gender-specific factors in youth substance use and mental health. This will help develop more effective, context-specific interventions that address the structural and psychosocial realities of urban informal settlements.

### Conclusions

The pilot study underscores the need to address the intertwined challenge of substance use and mental health among youth living in Nairobi's informal settlements. Within this sample, distinct risk patterns emerged across substances. Tobacco use was associated with higher depression and anxiety symptoms and lower educational attainment, suggesting that both psychological distress and limited schooling increase vulnerability. Alcohol use was associated with older age, greater depressive symptoms and independent living or employment, indicating how increased autonomy and income may facilitate access. These exploratory findings emphasize the importance of developing integrated, multilevel interventions that combine school retention

programs, youth employment initiatives, accessible mental health care and community-based harm reduction services. Such interventions must be developed in collaboration with young people and grounded in their lived realities to ensure feasibility and sustainability. As a pilot study, these findings generate several testable research questions for future work. First, how do structural factors, such as unemployment, housing instability and access to services, influence ASU trajectories among adolescents and young adults in informal settlements? Second, how does co-occurring psychological distress impact the onset, escalation, or maintenance of ASU behaviors? Third, which multilevel interventions that integrate community, family and individual components are most effective in reducing ASU and improving mental health outcomes? Longitudinal and intervention research is needed to clarify causal mechanisms and to evaluate multisectoral strategies aimed at improving health and well-being among urban youth in low-resource settings.

**Open peer review.** To view the open peer review materials for this article, please visit http://doi.org/10.1017/gmh.2025.10102.

**Data availability statement.** The data presented in this study are available on request from the corresponding author.

**Acknowledgments.** We thank the research participants, RAs and community collaborators for their involvement in this research. We are grateful to the Consortium for Advanced Research Training in Africa (CARTA) for financial support.

**Author contribution.** Conceptualization, C.M.M. and W.B.; formal analysis, W.B.; funding acquisition, C.M.M.; methodology, C.M.M., M.M. and W.B.; project administration, C.M.M. and T.M.; supervision, C.M.M., M.M., T.M. and A.M.; writing – original draft, C.M.M. and W.B.; writing – review and editing, T.M., A.M, D.N.K., S.M., M.M. and A.M. All authors have read and agreed to the published version of the manuscript.

**Financial support.** CMM received a reentry research award supported by the Consortium for Advanced Research Training in Africa (CARTA). CARTA is jointly led by the African Population and Health Research Center and the University of the Witwatersrand and funded by the Carnegie Corporation of New York (Grant No: B 8606.R02), SIDA (Grant No: 54100113), the DELTAS Africa Initiative (Grant No: 107768/Z/15/Z) and Deutscher Akademischer Austauschdienst (DAAD). The DELTAS Africa Initiative is an independent funding scheme of the African Academy of Sciences (AAS)'s Alliance for Accelerating Excellence in Science in Africa (AESA) and supported by the New Partnership for Africa's Development Planning and Coordinating Agency (NEPAD Agency) with funding from the Wellcome Trust (UK) and the UK government. The statements made and views expressed are solely the responsibility of the Fellow. The funders had no role in study design, data collection and analysis, decision to publish, or preparation of the manuscript.

**Competing interests.** The authors declare none.

**Ethics statement.** The study received ethical approval from the Kenyatta National Hospital and University of Nairobi (KNH-UoN ERC) No. P423/05/2024 and the National Commission for Science, Technology & Innovation (NACOSTI) No. NACOSTI/P/24/39331. Informed written consent/assent was obtained from participants before data collection commenced. For participants younger than 18, consent to participate was obtained from their parents or legal guardians.

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
