## [Reviewer Report]

This manuscript addresses a critical, under-researched issue and presents valuable baseline data. However, substantial methodological and analytic limitations, particularly regarding sample representativeness, confounding, context, and ethical protocols must be addressed.

1. Sampling and representativeness

The use of snowball sampling introduces substantial selection bias and limits the representativeness of the sample. The small sample size (n=94) drawn from only two settlements further constrains generalizability. Please expand the Limitations section to explicitly acknowledge how the sampling approach may have excluded the most marginalized youth and limited external validity.

In the Methods, describe in detail how recruitment via community health promoters (CHPs) and peer referral may have shaped the participant pool. Suggest, in the Discussion, that future research use probability sampling and broader recruitment across multiple settlements to improve generalizability.

2. Measurement and interpretation of substance Use

The current approach to estimating prevalence (summing “yes” responses) may overestimate polysubstance use and does not distinguish between experimental, social, regular, or dependent use. Please clarify in the measures section how substance use categories were operationalized, and specify any criteria or cut-offs used (e.g., based on ASSIST scoring). In the results section, provide more detail on the frequency, type, and severity of reported substance use. In the discussion section, acknowledge the risk of overestimation and discuss the implications for accurate prevalence and risk pattern interpretation.

3. PHQ-9 cultural validity

The manuscript does not address the validity of the PHQ-9 with Kenyan youth or those from informal settlements. Please indicate in the Measures section whether the PHQ-9 has been validated for this population or, if not, acknowledge this as a limitation.

Advise in the Discussion that future studies assess the PHQ-9’s cultural relevance and its ability to capture locally salient symptoms, including somatic or culture-specific expressions of distress.

4. Analytic approach

Analyses are limited to bivariate statistics, with no adjustment for potential confounders such as age, gender, or socioeconomic status. Please justify the analytic approach in the methods and explicitly acknowledge in the Results and Discussion the limitations of unadjusted analyses. If feasible, conduct and report multivariate analyses to provide adjusted findings; otherwise, clearly state the limitations of bivariate-only analyses for inference.

5. Interpretation of causality

Statements in the manuscript sometimes imply causality despite the cross-sectional study design. Please revise the discussion to avoid causal language, use “associated with” rather than “leads to” and ensure all wording is consistent with correlational data.

Include in the Limitations a clear statement that causal direction cannot be established from the current design.

6. Contextualization and practical implications

The Discussion does not fully explore how structural factors (poverty, unemployment, housing conditions) interact with substance use and mental health outcomes. Please expand the discussion to analyze these structural influences in greater depth.

In the conclusions, specify concrete, context-appropriate interventions (e.g., school-based programs, community outreach, integration with primary care, harm reduction measures), and discuss potential barriers to implementation and strategies to address them.

7. Referral for high-risk youth

The manuscript does not describe any procedures for referral or support for participants who screened positive for severe symptoms (depression or substance use). Please add a section in procedures/ethics outlining referral pathways and crisis protocols for high-risk participants, including any support mechanisms implemented. In the discussion, emphasize the ethical imperative for follow-up and support when conducting research with vulnerable populations.

8. Language and technical edits

Multiple instances of awkward phrasing, typographical errors, and inconsistent terminology are present throughout the manuscript.

Please carefully proofread the entire text to ensure clarity and professionalism. Standardize the use of terms (e.g., “substance use,” “alcohol and drug misuse,” “illicit drug use”) and ensure terms are defined operationally and used consistently throughout.

If reporting on “use” versus “misuse,” provide explicit criteria for this distinction. Check that all tables and figures are clearly labeled, consistently referenced in the text, and fully self-explanatory with appropriate titles and legends.

9. Demographic Reporting

The current reporting of participant demographics is limited. Please provide a more detailed breakdown in the results and tables, including age subgroups, school attendance, and living arrangement details. Ensure all tables/figures are comprehensive, self-contained, and appropriately referenced.

With revisions addressing the concerns above, this manuscript will make a valuable contribution to the regional and global literature on youth mental health in resource-constrained urban settings. To fully realize its potential, the manuscript requires considerable strengthening in methodological transparency, contextual interpretation, and ethical articulation.

---

## [Reviewer Report]

REVIEW: Epidemiology of Alcohol Misuse, Illicit Drug Use and Associated Mental Health Symptoms Among Youth in Informal Settlements of Nairobi, Kenya

This study addresses a critical and under-researched topic—substance use and mental health among youth in informal settlements in Kenya. The focus is timely and important, and the community-based approach is a strength. However, several areas require clarification and refinement to strengthen the rigor and interpretation of findings.

In the methods, please provide more detail on how eligibility was assessed, how youth with elevated symptoms were handled, and how confidentiality was maintained—particularly with minors. Additional detail is needed on the recruitment process, including potential biases introduced by how youth were “known,” RA training, and the handling of sensitive information. It would also strengthen the manuscript to clarify the use, adaptation, and validation of measures in this context, including whether any non-Western or locally developed tools were used.

In the results and discussion, the snowball sampling method limits generalizability, so the findings should be framed as exploratory. I recommend tempering conclusions about prevalence or group comparisons and emphasizing patterns within this specific sample of youth who use substances and perhaps changing the framing of this as an epi study of these things but more of a study of these specific youth.

Methods

- How was eligibility assessed? What happened with youth who had elevated symptoms? Were they referred somewhere for services?

- Provide brief sentences on ra level of training/education

- It says “youth were known” who knew them how, how might this bias the sample and change conclusions about this specific question or population

o More detail needed on this process

- Did RAs any specific training on interviewing or asking sensitive questions with youth

- How was confidentiality maintained with parents, imagine youth do not want to disclose SUD to parents, how was this navigated?

- Please provide more details in the measures about how or if measures have been validated or previously used in Kenya; whether Swahili versions were available, if there was adaptation or translation describe that

- Were any non-Western developed measures used or added in complement (i.e., measure or assessment of contextually related conceptions of stress or specific use of drugs/alcohol more often used in the area)

- What was randomized? When and how? You say mixed methods but there does not appear to be any qualitative component.

Results/Discussion

- Given the snowball sampling approach, the study’s findings should be framed as exploratory. I recommend tempering conclusions about group differences and prevalence, and clarifying that statistical tests highlight patterns within the sample rather than support generalizable or causal inferences.

- Sampling approach give us a picture of kids who are using so we can’t say there numbers reflect what we would find in any informal settlement in Kenya but more patterns of use and demo among youth who likely take in these settings

---

## [Editor Report]

Thank you for submitting your manuscript to Global Mental Health. As both reviewers noted, your paper addresses an important and under-researched topic. The reviewers raised several questions and concerns, primarily related to the methodology. We encourage you to revise the manuscript in line with their suggestions and to clarify the methodological issues and potential limitations they identified.

---

## [Reviewer Report]

The overall recommendation for the manuscript is Major Revisions Required.

The core reasoning for this recommendation is that while the study is a valuable contribution, its methodological limitations are significant enough to warrant a fundamental reframing of the paper. The study’s small sample size (n=94) and the use of non-probability snowball sampling introduce selection bias, which means the findings, such as the reported prevalence rates, cannot be confidently generalized to the wider population of youth in Nairobi’s informal settlements. Furthermore, the cross-sectional design and the use of only bivariate analyses prevent any conclusions about causality or the influence of confounding variables.

To address these issues and enhance the manuscript’s suitability for publication, the following revisions are recommended:

• Reframe the Manuscript: The paper should be presented as a “Pilot Study” or “Exploratory Research” rather than a standard “Research Article.” This change will align its claims with its actual methodological scope.

• Strengthen the Abstract: The abstract needs to be revised to explicitly state the exploratory nature of the work. The prevalence rates should be qualified as being specific to this sample, not as generalizable population estimates.

• Enhance the Discussion of Bias: A more detailed discussion of the likely direction of the selection bias is needed. The authors should explicitly state that the sample may over-represent individuals with substance use who are more socially connected.

• Refine Data Presentation: While the decision to use bivariate analysis is a sound one given the sample size, the authors could consider adding stratified analyses to provide a more nuanced understanding of the associations found.

• Expand Future Research Recommendations: The authors should elaborate on the specific research questions that this pilot study has generated to further cement its value as a foundational, hypothesis-generating work.

In conclusion, a Major Revision is not a rejection of the research but a clear directive to reframe the manuscript to align with the limitations of the data. This will ensure the paper’s scientific integrity and allow it to be published as a valuable piece of hypothesis-generating research, providing a strong and ethically sound foundation for future, larger-scale studies.

---

## [Editor Report]

Thank you for your thorough revision of the manuscript in response to the reviewers’ comments. One reviewer has recommended additional revisions to address the limitations introduced by the sampling approach and overall sample size, as well as the inferences that can reasonably be drawn from these data. I encourage the authors to carefully consider these points, particularly those related to potential selection bias and the interpretation of results. In particular, I agree with the reviewer that the findings should not be interpreted as prevalence estimates for the target or source population. Rather, it would be more appropriate to report the proportions of specific characteristics within the study sample.

---

## [Editor Report]

Thank you for submitting the revised draft of your manuscript, which has adequately addressed reviewer comments. Congratulations on this important article.